### Subject Category:
Chemistry

### Subject Areas:
biochemistry/environmental science

### Keywords:
simultaneous partial nitrification, anammox and denitrification, aerobic ammonia-oxidizing bacteria, anaerobic ammonia-oxidizing bacteria, denitrifying bacteria, membrane fouling

### Author for correspondence:
Jun Li
e-mail: bjutlj@163.com

This article has been edited by the Royal Society of Chemistry, including the commissioning, peer review process and editorial aspects up to the point of acceptance.

[†]The first two authors contributed equally to this work.

# Optimization of nitrogen and carbon removal with simultaneous partial nitrification, anammox and denitrification in membrane bioreactor

Kai Zhang[1,†], Zhaozhao Wang[2,†], Mengxia Sun[1], Dongbo Liang[1], Liangang Hou[2], Jing Zhang[1], Xiujie Wang[1] and Jun Li[1]

[1]National Engineering Laboratory of Urban Sewage Advanced Treatment and Resource Utilization Technology, The College of Architecture and Civil Engineering, Beijing University of Technology, Beijing 100124, People's Republic of China
[2]College of Energy and Environmental Engineering, Hebei University of Engineering, Handan 056038, People's Republic of China

JL, 0000-0002-3319-7503

In this study, a membrane bioreactor (MBR) was used to achieve both nitrogen and carbon removal by a simultaneous partial nitrification, anammox and denitrification (SNAD) process. During the entire experiment, the intermittent aeration (non-aerobic time : aeration time, $\text{min min}^{-1}$) cycle was controlled by a time-controlled switch, and the aeration rate was controlled by a gas flowmeter, and the optimal operating parameters as determined by response surface methodology (RSM) were a C/N value of 1.16, a DO value of $0.84 \text{ mg l}^{-1}$ and an aerobic time ($T_{ae}$) of 15.75 min. Under these conditions, the SNAD process achieved efficient and stable nitrogen and carbon removal; the total inorganic nitrogen removal efficiency and chemical oxygen demand removal efficiency were 92.31% and 95.67%, respectively. With the formation of granular sludge, the membrane fouling rate decreased significantly from $35.0 \text{ Pa h}^{-1}$ at SNAD start-up to $19.9 \text{ Pa h}^{-1}$ during stable operation. Fluorescence *in situ* hybrid analyses confirmed the structural characteristics and the relative ratio of aerobic ammonia-oxidizing bacteria, anaerobic ammonia-oxidizing bacteria and denitrifying bacteria in the SNAD system.

# 1. Introduction

Conventional biological nitrogen removal occurs via nitrification and denitrification (N/DN) through the synergistic action of ammonia-oxidizing bacteria (AOB), nitrite-oxidizing bacteria (NOB) and denitrifying bacteria (DNB) [1]. However, conventional biological nitrogen removal via the N/DN process has obvious drawbacks, including the requirement of a large area, substantial aeration costs and the need for an additional carbon source [2]. Anaerobic ammonium oxidation (anammox) has been confirmed as one of the most cost-effective nitrogen removal processes [3]. Anammox bacteria use nitrite as an electron acceptor to oxidize ammonium into nitrogen, and nitrate is formed in the process of the reaction [4]. According to the stoichiometric formula, 0.26 mol nitrate is produced when 1 mol ammonium and 1.32 mol nitrite are consumed in the process of anammox [5]. Nitrate is mainly produced by anaerobic ammonia-oxidizing bacteria (AnAOB) and NOB, and the production of excessive nitrate directly affects the removal rate of total nitrogen [6].

The simultaneous partial nitrification, anammox and denitrification (SNAD) process has recently been widely researched because of its synchronous nitrogen and carbon removal [7]. DNB are cultured in the SNAD system to remove nitrate during the simultaneous removal of carbon by denitrification. The advantage of the SNAD process is that nitrogen removal and chemical oxygen demand (COD) reduction are completed in a single system [8]. Three main biochemical reactions are involved in the SNAD process. Under the condition of oxygen limitation, partial ammonium is oxidized to nitrite by AOB (equation (1.1)), and then the remaining ammonium and the formed nitrite act as the substrate of AnAOB to form nitrate and dinitrogen gas (equation (1.2)). Finally, the residue of nitrite and the produced nitrate are reduced to dinitrogen gas through the denitrifying process (equations (1.3–1.5)). Therefore, using intermittent aeration mode, SNAD can simultaneously remove nitrogen and carbon in a single reactor [9].

$$NH_4^+ + 1.5O_2 \rightarrow NO_2^- + H_2O + 2H^+, \tag{1.1}$$

$$NH_4^+ + 1.31NO_2^- + 0.066HCO_3^- + 0.13H^+ \rightarrow 1.02N_2 + 0.26NO_3^- + 0.066CH_2O_{0.5}N_{0.15} + 2.03H_2O, \tag{1.2}$$

$$NO_3^- + 0.675CH_3COOH \rightarrow 0.5HCO_3^- + 0.5N_2 + 0.125CO_2 + 0.625H_2O, \tag{1.3}$$

$$NO_2^- + 0.375CH_3COOH \rightarrow 0.5HCO_3^- + 0.5N_2 + 0.25CO_2 + 0.5H_2O, \tag{1.4}$$

and

$$NO_3^- + 0.25CH_3COOH \rightarrow NO_2^- + 0.5CO_2 + 0.5H_2O. \tag{1.5}$$

To date, there have been some reports about the SNAD process. Some studies have found that the existence of COD inhibits both aerobic ammonia-oxidizing bacteria (AerAOB) and AnAOB [10]. It was also reported that a maximum total nitrogen removal efficiency of 97.47% was obtained at a C/N ratio of 2, and that the activity of AnAOB was inhibited under the condition of high organic matter concentration, which led to the decrease in total nitrogen removal rate [11]. However, anammox can quickly restore activity after inhibition. Tang *et al.* [12] demonstrated that AnAOB growth was significantly suppressed by DNB under the influence of a $COD/NO_2^- - N$ ratio of 2.92. In addition, it has been reported that the addition of a carbon source can promote the growth of heterotrophic bacteria, which secrete a large amount of EPS and bind to nitrifying bacteria, thus reducing the loss of nitrifying bacteria. This is conducive to the stable accumulation of nitrite nitrogen and providing a substrate for AnAOB [13]. The factors that affect the stable operation of the SNAD process include DO, C/N and particle size [14]. However, most of the previous related studies have focused on the influence of a single factor and ignored the roles of other factors. Thus, the findings of different researchers vary greatly, resulting in one-sided conclusions. Therefore, the effects of DO, C/N and particle size on the simultaneous removal of nitrogen and carbon in the SNAD system were investigated in the course of the present study.

In this study, a membrane bioreactor (MBR) was applied for the SNAD process. The MBR completely intercepted the slow-growing AnAOB in the reactor, and maintained high-concentration microbial biomass [8]. Some studies have demonstrated the superiority of combining MBR with AnAOB, especially from the aspects of TN removal performance and the reduction in membrane fouling [15–17]. However, MBR systems often suffer from severe membrane fouling due to the deposition of EPS and flocculent sludge on the surface of the membrane [1]. Previous scholars have conducted a significant amount of research on mitigating membrane fouling, such as by using $TiO_2$ nanomaterials to change the surface characteristics of the membrane, and increasing the shear force on the membrane surface by $N_2$ to reduce the membrane fouling [18]. At present, EPS is considered to be the

**Table 1.** Operational periods of the experiment.

| phase | influent concentration (mg l$^{-1}$) | | | $T_{an}:T_{ae}$ (min) | DO (mg l$^{-1}$) | aeration rate (ml min$^{-1}$) | $T$ (°C) | HRT (h) |
| --- | --- | --- | --- | --- | --- | --- | --- | --- |
| | COD | $NH_4^+$—N | $NO_2^-$—N | | | | | |
| I (1–55 days) | 100 | 200 | 20 | 7 : 15 | 0–0.8 | 400 | 30 ± 2 | 8 |
| II (56–114 days) | 50–400 | 200 | 20 | 7 : 7 | 0–0.8 | 400–700 | 30 ± 2 | 8 |
| III (115–160 days) | 200 | 200 | 20 | 7 : (7–23) | 0–0.8 | 200–400 | 30 ± 2 | 8 |
| IV (161–203 days) | 200 | 200 | 20 | 7 : 15 | 0.3–0.8 | 500–700 | 30 ± 2 | 8 |

most critical factor that affects membrane fouling [1]. However, EPS secreted by AnAOB cannot only be used as a carbon source, but also as a binder to promote cell agglomeration [19]. Therefore, with the increase in sludge particle size, the sedimentation performance of granular sludge is improved and the accumulation of granular sludge on the membrane surface is reduced to a certain extent under the action of a stirrer, ultimately reducing the rate of membrane fouling.

In this study, response surface methodology (RSM) was applied to optimize the operating parameters. The effects of different variables (factors) on different responses (results) can be simulated by RSM. The objectives of the present study were (i) to establish a stable SNAD process and investigate the activity of functional bacteria and the removal rates of nitrogen and carbon at different stages and (ii) to investigate the biological community structure and membrane fouling characteristics. The results of this study will provide a theoretical basis for understanding the variation between the operating parameters and biological structure in the SNAD process.

# 2. Material and methods

## 2.1. Experimental set-up and operation

A bench-scale MBR-SNAD reactor with a working volume of 53.5 l was used. A membrane module (hollow fibre PVDF membrane) was installed in the reactor. The length of the membrane module was 33 cm, the outer diameter of the membrane filament was 3 mm, the pore diameter of the membrane was 0.03 μm and the effective filtration area was 0.519 m$^2$ (Hangzhou Microna Membrane Technology Co., Ltd, China). The bottom of the reactor was provided with an aeration disc. The intermittent aeration cycle was controlled by a time-controlled switch, and the aeration rate was controlled by a gas flowmeter. To improve the substrate transfer ability, the agitator worked at a constant speed (80 r min$^{-1}$) throughout the operation of the reactor. The reactor operated at 30 ± 2°C, and was equipped with a water bath. A black soft material was used to avoid light in the reactor. During the operation of the reactor, with the increase in the transmembrane pressure (TMP), the speed of the peristaltic pump was increased correspondingly to ensure a constant effluent. When the TMP reached 50 kPa, the membrane components were immersed in a solution of sodium hypochlorite for chemical cleaning. The SRT was maintained for 50 days. The operation of the reactor is summarized in table 1 and electronic supplementary material, figure S3.

In this study, the synthetic wastewater that was used contained NH$_4$Cl, CH$_3$COONa and NaHCO$_3$ as the main substrates. Additionally, small amounts of KH$_2$PO$_4$, CaCl$_2$, MgSO$_4$·7H$_2$O and trace element solution (1 ml l$^{-1}$) were added. The trace element solution contained the following (g l$^{-1}$): 15.00 EDTA, 0.43 ZnSO$_4$·4H$_2$O, 0.99 MnCl$_2$·4H$_2$O, 0.014 H$_3$BO$_4$, 0.25 CuSO$_4$·5H$_2$O, 0.22 Na$_2$MoO$_4$·2H$_2$O, 0.21 Na$_2$SeO$_4$·10H$_2$O and 0.19 NiCl$_2$·6H$_2$O.

## 2.2. Seed sludge

The inoculated sludge included activated sludge from Handan West sewage treatment plant and anammox floc sludge from the stable UASB reactor in the laboratory of Beijing University of

Technology. In the MBR, activated sludge was firstly inoculated for domesticating and enriching AOB to achieve stable short-cut nitrification, and then anaerobic ammoxidation flocculent sludge was added to achieve stable operation of the CANON process. In this study, the start-up and operation of the SNAD process was conducted on the basis of the above 204 days of experiments.

## 2.3. Analytical methods

$NH_4^+-N$, $NO_2^--N$, $NO_3^--N$, COD and a mixed liquor suspended solid (MLSS) were analysed according to standard methods [20]. Influent and effluent samples were collected every 2 or 3 days during the reactor operation. The DO concentration, pH and temperature were determined with oxygen, pH and temperature probes (WTW 3420, Germany). The sludge particle size distribution (PSD) was determined by the sieving method using a series of $\Phi$ 10 cm stainless steel to determine the scope of the sludge particle size, and followed the principle of limit gauge with particles that could move through the standard screen to determine the particle size. The concentrations of $\rho(KH_2PO_4) = 4$ g $l^{-1}$, $\rho(Na_2HPO_4 \cdot 7H_2O) = 5$ g $l^{-1}$, and $\rho(KH_2PO_4) = 1.2$ g $l^{-1}$ and pH = 7.1 provided sufficient osmotic pressure and pH for the particles of the buffer solution to ensure that the structure of the granular sludge was not destroyed. After, the particles were grouped into classes by size corresponding to the sieving (less than 0.38, 0.061, 0.1, 0.154, 0.2, 0.335, 0.6, 1 and greater than 2 mm) and then the total suspended solid (TSS) was calculated for each class [21]. The membrane flux was determined by equation (2.1) [16], and the membrane fouling rate (MFR, Pa $h^{-1}$) was determined by equation (2.2) [22]

$$J = \frac{V}{(A \times t)},$$ (2.1)

where $V$ is the volume of permeate liquid at time $t$ (l), $A$ is the effective area of the membrane ($m^2$) and $t$ is the time (h)

$$MFR = \frac{dTMP}{dt} = \frac{TMP_{end} - TMP_{initial}}{t_{end} - t_{initial}},$$ (2.2)

where $TMP_{end}$ is the transmembrane pressure at the end (Pa), $TMP_{initial}$ is the initial transmembrane pressure (Pa) and $t$ is the time (h).

## 2.4. Microbial activity tests

The activities of AerAOB, AnAOB, $DNB_a$ and $DNB_b$ were tested during the stable operation of the SNAD process. The batch experiment conditions are described in electronic supplementary material, table S1. Before the batch test, the activated sludge was washed three times, and 15 g of wet sludge was then injected into 500 ml beakers to test the microbial activity. Finally, the total volume of adding 15 g wet sludge and medium was 500 ml. In addition, the temperature was maintained at $30 \pm 2°C$. All tests were conducted in triplicate.

## 2.5. Response surface methodology

During the experimental design process, C/N, DO and $T_{ae}$ were selected as the three independent effective variables. Each variable was evaluated at three levels according to a previous study [8]. The C/N, DO and $T_{ae}$ were denoted as $X_1$, $X_2$ and $X_3$, respectively. The experimental conditions and results are listed in electronic supplementary material, table S2. The three factors were modelled according to the below equation

$$Y_i = \beta_0 + \sum_{i=1}^{k} \beta_i X_i + \sum_{i=1}^{k} \beta_{ii} X_i^2 + \sum_{i<1}^{k} \beta_{ij} X_i X_j + \varepsilon,$$ (2.3)

where $Y_i$ is the response (including the TIN removal efficiency and COD removal efficiency), $X_i$ represents the independent variable, $\beta_0$ is a constant, $\beta_i$, $\beta_{ii}$ and $\beta_{ij}$ are the parameters of the regression model, $k$ is the number of variables and $\varepsilon$ is the random error. In addition, Design-Expert software was employed to generate the response surface and contour plots, and the models were analysed via analysis of variance (ANOVA).

**Table 2.** Oligonucleotide probes used for FISH.

| probe name | target organism | probe sequence (50–30) | labelling dye |
|---|---|---|---|
| EUB338 | ALL bacteria | GCT GCC TCCCGT AGG AGT; | FAM |
| EUB338 II | | GCA GCC ACCCGT AGG TGT; | |
| EUB338 III | | GCT GCC ACCCGT AGG TGT; | |
| NOS190 | AOB | CGA TCC CCTGCT TTT CTCC | Cy3 |
| NTSPA662 | NOB | GGA ATT CCGCGC TCC TCT | FITC |
| AMX820 | anammox | AAAACCCCTCTACTTAGTGCCC | Cy3 |
| PDV198 | denitrifying bacteria | CTAATCCTTTGGCGATAAATC | FITC |

## 2.6. Fourier transform infrared spectroscopy analysis

FT-IR was used to analyse the functional groups contained in the pollutants on the membrane surface to determine the main composition of the pollutants. The fouling material on the surface of the film was ground with KBr in the proportion of 1 : 10, and transmission scanning was then conducted with an infrared spectrometer after forming the pressed film. Origin8.0 software was used to analyse the collected data.

## 2.7. Scanning electron microscope observation

The morphology and structural characteristics of the biomass during the stable operation of the SNAD process were observed by scanning electron microscope (SEM) [22]. The samples were supplemented with 2.5% glutaraldehyde with a pH of 6.8, placed in a refrigerator at 4°C for 1.5 h for fixation, and dehydrated with ethanol at a concentration of 50–100% in 10% increments. The samples were then freeze-dried for 12 h and covered with a palladium/gold alloy. Finally, the samples were analysed with SEM (JEOL, JSM-7800 F, Tokyo, Japan).

## 2.8. Fluorescence *in situ* hybrid analysis

Fluorescence *in situ* hybrid (FISH) analysis was used to observe the coexistence of AerAOB, anaerobic ammonia-oxidizing bacteria (AnAOB) and denitrification bacteria (DNB). Cell fixation and FISH analysis were conducted according to the standard protocol [23]. The probes (described in table 2) were obtained from Sangon (Shanghai, China), and all the probes were complementary to regions of the small subunit of 16S rRNA molecules and conjugated with fluorescent dyes. Hybridizations were performed on 4% (w/v) paraformaldehyde-fixed samples for 4 h, which were then stored in phosphate-buffered saline/ethanol (1 : 1) at −20°C for further processing [24]. The hybridized samples were observed with a confocal laser scanning microscope (FV1200, Olympus, Japan). The population in the selected area was quantitatively analysed by Image Pro-Plus 6.0 software.

# 3. Results and discussion

## 3.1. Removal performances of nitrogen and carbon in the SNAD process

The nitrogen and COD performances of the reactor are presented in figure 1. The reactor was operated for 203 days. The temperature and pH were maintained at 32°C and 7.8–8.0, respectively. As shown in figure 1, the experiment period can be divided into four stages: the start-up of the SNAD period (Phase I, days 1–55), and the SNAD process optimization period (Phase II, days 56–114; Phase III, days 115–160; Phase IV, days 161–203).

In Phase I, the influent $NH_4^+-N$ and $NO_2^--N$ concentrations were 200 mg l$^{-1}$ and 20 mg l$^{-1}$, respectively. During this phase, COD (100 mg l$^{-1}$) was introduced into the influent synthetic wastewater to start the SNAD process. As shown in figure 1a, during the first 13 days, the $NH_4^+-N$ removal efficiency and the TIN removal efficiency were less than 70% and 60%, respectively. This might be due to the inhibitory effect of COD on anammox, which, therefore, required more time to

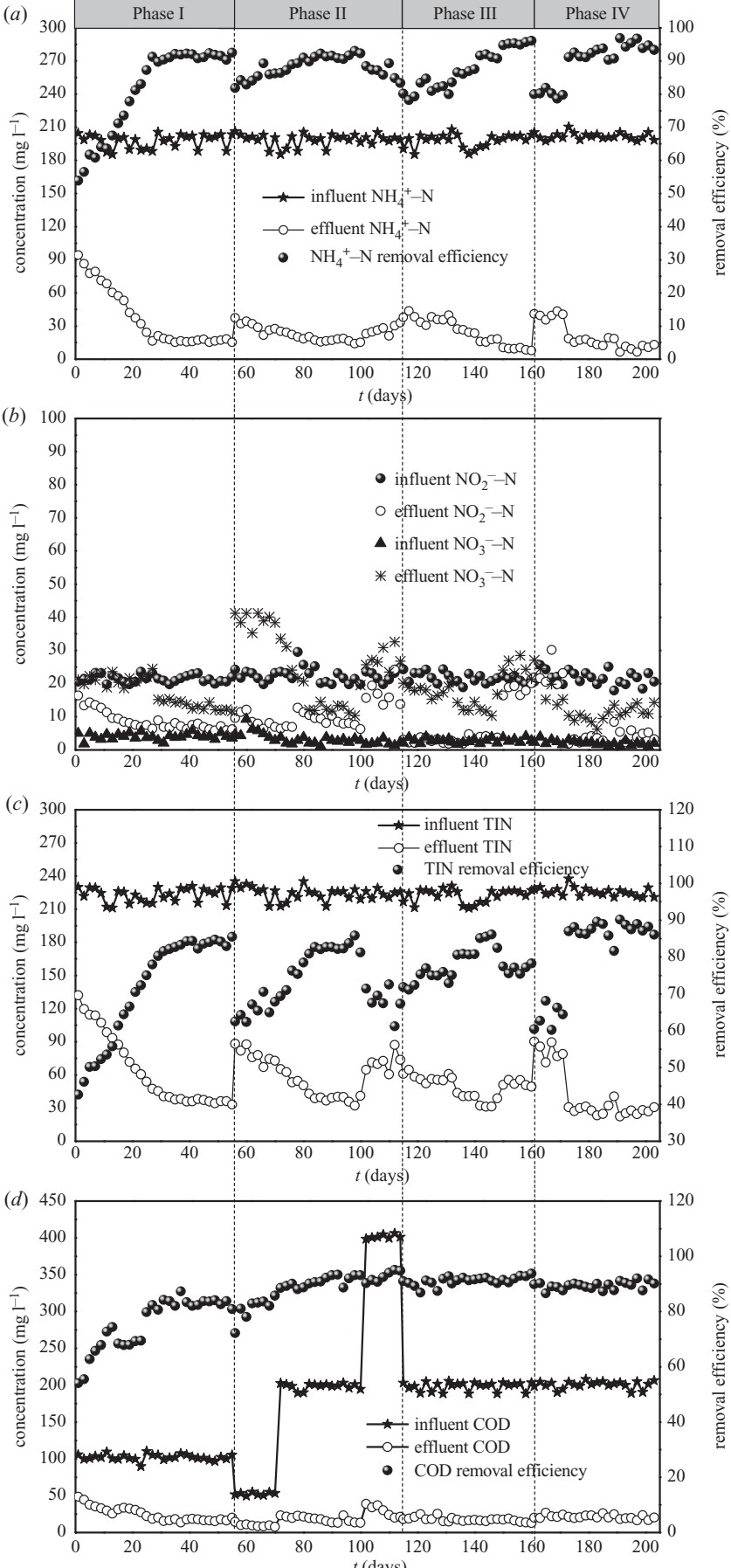

**Figure 1.** The nitrogen and COD removal performance of the reactor.

adapt to the new system. From days 14 to 29, the effluent $NH_4^+-N$, TIN and COD concentrations decreased from 57.5, 87.3 and 31.5 mg l$^{-1}$ to 21.2, 45.2 and 20.4 mg l$^{-1}$, respectively. Correspondingly, the $NH_4^+-N$, TIN and COD removal efficiencies increased from 71.2%, 61.3% and 68.4% to 89.7%, 80.3% and 80.6%, respectively (figure 1a,c,d). Afterwards, the system gradually achieved stable performance (from days 30 to 55). During these days, the average effluent $NH_4^+-N$, $NO_2^--N$, $NO_3^--N$, TIN and COD concentrations were 16.7, 6.8, 13.3, 36.9 and 16.8 mg l$^{-1}$, respectively (figure 1). Correspondingly, the average $NH_4^+-N$, TIN and COD removal efficiencies reached 91.5%, 83.5% and 83.4%, respectively.

In Phase II, the C/N ratio was gradually increased from 0.25 to 2, and the intermittent aeration mode (non-aeration time to aeration time (min : min)) and the DO were maintained at 7 : 7 and 0.8, respectively. As shown in figure 1, from days 56 to 70, the average influent concentrations of COD, $NH_4^+-N$ and $NO_2^--N$ were 52.3, 200.0 and 22.3 mg l$^{-1}$, respectively. During these days, the average effluent $NO_3^--N$ concentration was greater than 40 mg l$^{-1}$, and the removal efficiencies of $NH_4^+-N$ and TIN decreased from 92.4% and 85.5% at the end of Phase I to 86.2% and 67.8%, respectively. In a study by Wang et al. [8], it was found that when the C/N ratio was 0.33, the removal efficiency was only 88.7%, mainly because it could not provide sufficient COD to remove the carbon source in the SNAD process. The results of another previous study [25] revealed that with the increase in COD, the $NH_4^+-N$ removal efficiency decreased gradually, and the addition of COD could inhibit the activity of anammox and lead to the decrease in TIN removal efficiency, regardless of COD concentration. From days 71 to 100, the COD concentration increased to 200 mg l$^{-1}$ (C/N = 1), and the $NH_4^+-N$, TIN and COD removal efficiencies increased from 86.4%, 69.3%, and 88.5% to 92.2%, 81.3% and 93.2%, respectively (figure 1a,c,d). At the same time, the effluent $NO_3^--N$ concentration decreased from 33.6 to 12.5 mg l$^{-1}$ (figure 1b). Guo et al. [26] reported that the oxygen saturation constant ($k_0$) of nitrate-oxidizing bacteria (NOB) was higher than that of AOB. Hence, in this study, the activity of NOB was more obviously affected than that of AOB under the conditions of a low DO and low intermittent aeration mode. From days 101 to 114, the influent COD concentration further increased to 400 mg l$^{-1}$ (C/N = 2), resulting in the reduction in the $NH_4^+-N$ and TIN removal efficiencies from 89.3% and 82.3% to 83.3% and 67.3%, respectively (figure 1a,c). However, the removal efficiency of COD was still greater than 90% (figure 1d). The results demonstrate that with the increase in COD, heterotrophic DNB were superior to autotrophic anammox microbes in the competition for the substrate ($NO_2^--N$) and living space. Some studies have shown that the existence of COD is not suitable for the growth of anammox, so the process based on anammox is suitable for the treatment of wastewater with a low C/N ratio [27,28].

In Phase III, the influent $NH_4^+-N$ concentration was 200 mg l$^{-1}$, and the C/N ratio and DO were 1 and 0.8 mg l$^{-1}$, respectively. The non-aeration time was maintained at 7 min, and the aeration time was gradually increased from 7 to 23 min. From days 115 to 132, the aeration time was 7 min. As shown in figure 1, the average effluent $NH_4^+-N$ concentration was 36.8 mg l$^{-1}$. Correspondingly, the average $NH_4^+-N$ removal efficiency was 81.4%. These results indicate that the removal efficiency of $NH_4^+-N$ was low due to the short aeration time and the lack of DO due to AerAOB. In addition, the average effluent $NO_2^--N$, $NO_3^--N$, TIN and COD concentrations were 2.7, 18.2, 57.7 and 19.4 mg l$^{-1}$, respectively (figure 1b,c,d). From days 133 to 148, the aeration time was extended to 15 min, and the effluent $NH_4^+-N$ concentration gradually decreased from 32.1 to 18.3 mg l$^{-1}$. Meanwhile, the removal efficiency of TIN increased from 78.6% to 82.5% (figure 1c). From days 149 to 160, the aeration time was prolonged to 23 min, the effluent $NH_4^+-N$ concentration decreased from 18.2 to 7.9 mg l$^{-1}$ and the effluent $NO_3^--N$ concentration increased from 38.5 to 49.4 mg l$^{-1}$. The long-term aeration enhanced the activity of NOB.

In Phase IV, the influent $NH_4^+-N$ and $NO_2^--N$ concentrations were 200 mg $\cdot$ l$^{-1}$ and 20 mg $\cdot$ l$^{-1}$, respectively, and the C/N ratio and aeration time were 1 min and 15 min, respectively. From days 161 to 203, the DO gradually increased from 0.3 to 0.8 mg l$^{-1}$. As shown in figure 1a, during the period in which the DO was 0.3 mg l$^{-1}$ (from days 161 to 171), the average effluent $NH_4^+-N$ concentration was 40.1 mg l$^{-1}$, and the average TIN removal efficiency was only 63.6% (figure 1a). From days 172 to 203, the DO of the aeration stage was 0.8 mg l$^{-1}$, the effluent $NH_4^+-N$ concentration gradually decreased to 12.4 mg l$^{-1}$, and the removal efficiency of TIN was stable above 85% (figure 1c). In the last stage (from days 188 to 203), the average effluent $NH_4^+-N$, $NO_2^--N$, $NO_3^--N$ and COD concentrations were 10.1, 4.3, 12.1 and 19.1 mg l$^{-1}$, respectively (figure 1a,d). Correspondingly, the average TIN and COD removal efficiencies were 88.2% and 90.4%, respectively. From days 172 to 203, the $NH4^+-N$ removal rate tends to increase with the increasing of DO, indicating that DO not only meets the requirements of heterotrophic bacteria, but also provides sufficient DO environment for AOB. In

**Table 3.** Analysis of TIN removal efficiency variance table. $\sqrt{}$, significant; $\times$, not significant.

| source | sum of squares | d.f. | mean squares | $F$-value | $p$-value | |
|---|---|---|---|---|---|---|
| model | 765.07 | 9 | 85.01 | 22.3 | 0.0002 | $\sqrt{}$ |
| $X_1$—C/N | 89.78 | 1 | 89.78 | 23.55 | 0.0019 | $\sqrt{}$ |
| $X_2$—DO | 40.05 | 1 | 40.05 | 10.51 | 0.0142 | $\times$ |
| $X_3$—$T_{ae}$ | 30.81 | 1 | 30.81 | 8.08 | 0.0249 | $\times$ |
| $X_1X_2$ | 25.5 | 1 | 25.5 | 6.69 | 0.0361 | $\times$ |
| $X_1X_3$ | 7.56 | 1 | 7.56 | 1.98 | 0.2018 | $\times$ |
| $X_2X_3$ | 1.21 | 1 | 1.21 | 0.32 | 0.5908 | $\times$ |
| $X_1^2$ | 64.37 | 1 | 64.37 | 16.89 | 0.0045 | $\sqrt{}$ |
| $X_2^2$ | 378.8 | 1 | 378.8 | 99.37 | <0.0001 | $\sqrt{}$ |
| $X_3^2$ | 79.13 | 1 | 79.13 | 20.76 | 0.0026 | $\sqrt{}$ |
| residual | 26.69 | 7 | 3.81 | | | |
| lack of fit | 7.48 | 3 | 2.49 | 0.52 | 0.6915 | $\times$ |

**Table 4.** Analysis of COD removal efficiency variance table. $\sqrt{}$, significant; $\times$, not significant.

| source | sum of squares | d.f. | mean squares | $F$-value | $p$-value | |
|---|---|---|---|---|---|---|
| model | 236.98 | 9 | 26.33 | 23.65 | 0.0002 | $\sqrt{}$ |
| $X_1$—C/N | 59.41 | 1 | 59.41 | 53.35 | 0.0002 | $\sqrt{}$ |
| $X_2$—DO | 0.55 | 1 | 0.55 | 0.5 | 0.5044 | $\times$ |
| $X_3$—$T_{ae}$ | 0.78 | 1 | 0.78 | 0.7 | 0.4299 | $\sqrt{}$ |
| $X_1X_2$ | 9.3 | 1 | 9.3 | 8.35 | 0.0233 | $\times$ |
| $X_1X_3$ | 0.023 | 1 | 0.023 | 0.02 | 0.891 | $\times$ |
| $X_2X_3$ | 6.76 | 1 | 6.76 | 6.07 | 0.0432 | $\times$ |
| $X_1^2$ | 85.07 | 1 | 85.07 | 76.4 | <0.0001 | $\sqrt{}$ |
| $X_2^2$ | 22.66 | 1 | 22.66 | 20.35 | 0.0028 | $\times$ |
| $X_3^2$ | 37.14 | 1 | 37.14 | 33.35 | 0.0007 | $\sqrt{}$ |
| residual | 7.79 | 7 | 1.11 | | | |
| lack of fit | 1.52 | 3 | 0.51 | 0.32 | 0.8095 | $\times$ |

addition, with the decrease in $NH_4^+$—N concentration and TIN concentration of effluent, anammox may occur in the reactor. In summary, the SNAD process operated stably in the intermittent aeration MBR system. In this study, three main biochemical reactions are involved in the SNAD process. Under the condition of intermittent aeration, partial ammonium is oxidized to nitrite by AOB, and then the remaining $NH_4^+$—N and the formed $NO_2^-$—N act as the substrate of AnAOB to form $NO_3^-$—N and $N_2$. Finally, the residue of $NO_2^-$—N and the produced $NO_3^-$—N are reduced to $N_2$ through the denitrifying process. Accordingly, how to optimize the intermittent aeration and C/N ratio becomes the difficulty of SNAD process stable operation.

## 3.2. Analysis of response surface methodology

As shown in equations (3.1–3.4), according to the removal efficiencies of TIN and COD under different C/N, DO and $T_{ae}$ conditions, RSM was used to establish the final equation of different responses in the SNAD system expressed by coding and actual factors. The experimental results are exhibited in electronic supplementary material, table S2. The results of the evaluation are presented in tables 3 and 4. For the

TIN removal efficiency model, $F = 22.30$ and $p = 0.0002$, indicating that the model used in the experiment was extremely significant. The corresponding items ($X_1$, $X_1^2$, $X_2^2$ and $X_3^2$) in the model all had $p < 0.005$, indicating that they had an extremely significant influence on the TIN removal efficiency. The predicted $R^2$ of 0.8110 was in reasonable agreement with the adjusted $R^2$ of 0.9230. The regression coefficient $R^2 = 0.9663$ indicates that the model had a good fitting degree

$$Y_1 = 90.62 - 4.35X_1 + 2.49X_2 + 1.46X_3 - 2.28X_1X_2$$
$$+ 2.12X_1X_3 - 1.08X_2X_3 - 5.99X_1^2 - 12.56X_2^2 - 7.91X_3^2, \tag{3.1}$$

$$TIN(\%) = 14.67 + 12.22X_1 + 97.95X_2 + 3.90X_3 - 5.20X_1X_2$$
$$+ 0.30X_1X_3 - 0.45X_2X_3 - 7.81X_1^2 - 50.24X_2^2 - 0.12X_3^2, \tag{3.2}$$

$$Y_2 = 93.66 + 3.17X_1 - 1.26X_2 - 1.61X_3 - 1.45X_1X_2$$
$$+ 0.45X_1X_3 - 1.12X_2X_3 - 4.34X_1^2 - 4.12X_2^2 - 3.17X_3^2, \tag{3.3}$$

$$COD(\%) = 60.49 + 18.07X_1 + 31.77X_2 + 1.43X_3 - 3.31X_1X_2$$
$$+ 0.06X_1X_3 - 0.28X_2X_3 - 5.67X_1^2 - 16.47X_2^2 - 0.04X_3^2. \tag{3.4}$$

As shown in figure 2$a$, the measured values of the experiment had a good linear relationship with the predicted values. In addition, the adequate precision was 12.40, which was greater than 4, thereby indicating that this model can be used to guide the experimental design. Therefore, the model has high reliability and can theoretically predict and evaluate the results of TIN removal efficiency. The COD removal efficiency model was significant with $F = 23.65$ and $p = 0.0002$. The corresponding items ($X_1$, $X_1^2$, $X_2^2$ and $X_3^2$) all had $p < 0.005$, which implies that the items had an extremely significant influence on the COD removal efficiency. Additionally, the approximate equality between the predicted $R^2 = 0.8604$ and the actual $R^2 = 0.9272$ prove the relative reliability of the model. The regression coefficient $R^2 = 0.9682$ indicates that the equation simulation was more accurate.

As shown in figure 2$e$, the measured values of the experiment had a good linear relationship with the predicted values. In addition, the adequate precision was 13.99, which was greater than 4, thereby implying that this model can be used to guide the experimental design. In summary, the model has high reliability and can theoretically predict and evaluate the results of COD removal efficiency.

In the figure, the $x$-axis and $y$-axis represent C/N, DO and $T_{ae}$, respectively. The $z$-axis represents the TIN removal efficiency or COD removal efficiency. The 3D response surface plot and the corresponding contour plot of the influence of the interaction of various factors on the removal efficiency of TIN are presented in figure 2$b$–$d$. Figure 2$b$ shows the influence of the C/N ratio and DO on the TIN removal efficiency under a constant aeration time ($T_{ae} = 15$ min). When the DO is between 0.3 and 1.3 mg l$^{-1}$, the TIN removal efficiency first increases and then decreases with the increase in the C/N ratio. The TIN removal efficiency first increases and then decreases with the increase in DO when the C/N ratio ranges from 0.25 to 2. When the DO remains constant, sufficient organic carbon sources are provided for DNB in the SNAD system with the increase in the C/N ratio, thus improving the TIN removal efficiency. However, the continued increase in organic carbon sources might inhibit AnAOB [29], leading to the decrement of the TIN removal efficiency. As shown in figure 2$b$, when the C/N ratio and DO are 0.57–0.82 and 0.85–0.94 mg l$^{-1}$, respectively, the TIN removal efficiency is greater than 93.1%. Figure 2$c$ illustrates the effect of the interaction between C/N and $T_{ae}$ on the TIN removal efficiency at a constant DO of 0.8 mg l$^{-1}$. When the C/N ratio and $T_{ae}$ are between 0.57 and 0.97 and 14.3 and 18.3 min, respectively, the TIN removal efficiency is greater than 92.8%. Figure 2$d$ demonstrates the combined effect of DO and $T_{ae}$ on the TIN removal efficiency at a constant C/N (1.13). The maximum removal efficiency of TIN occurs at a DO of 0.75–0.95 mg l$^{-1}$ and $T_{ae}$ of 14.5–19.1 min. Under the condition in which $T_{ae}$ remains unchanged, it has little effect on the TIN removal efficiency, whereas the DO has a significant effect on the TIN removal efficiency. This result indicates that DO is an important control parameter. The simultaneous effects of DO and C/N on the COD removal efficiency are depicted in figure 2$f$. When DO remains unchanged, the C/N ratio is increased from 0.25 to 1.36, and the DNB activity is gradually enhanced, thus promoting the removal of COD. However, when the C/N ratio ranges from 1.36 to 2.0, the COD removal efficiency gradually decreases, which may be due to the increase in the concentration of organic carbon sources, thus inhibiting the activities of AOB and AnAOB, and limiting the production of $NO_2^- - N$ and $NO_3^- - N$ as substrates for DNB. Figure 2$g$ shows the combined effect of the C/N ratio and $T_{ae}$ on the COD removal efficiency at a constant DO (0.8 mg l$^{-1}$). For a constant $T_{ae}$, the COD removal efficiency first

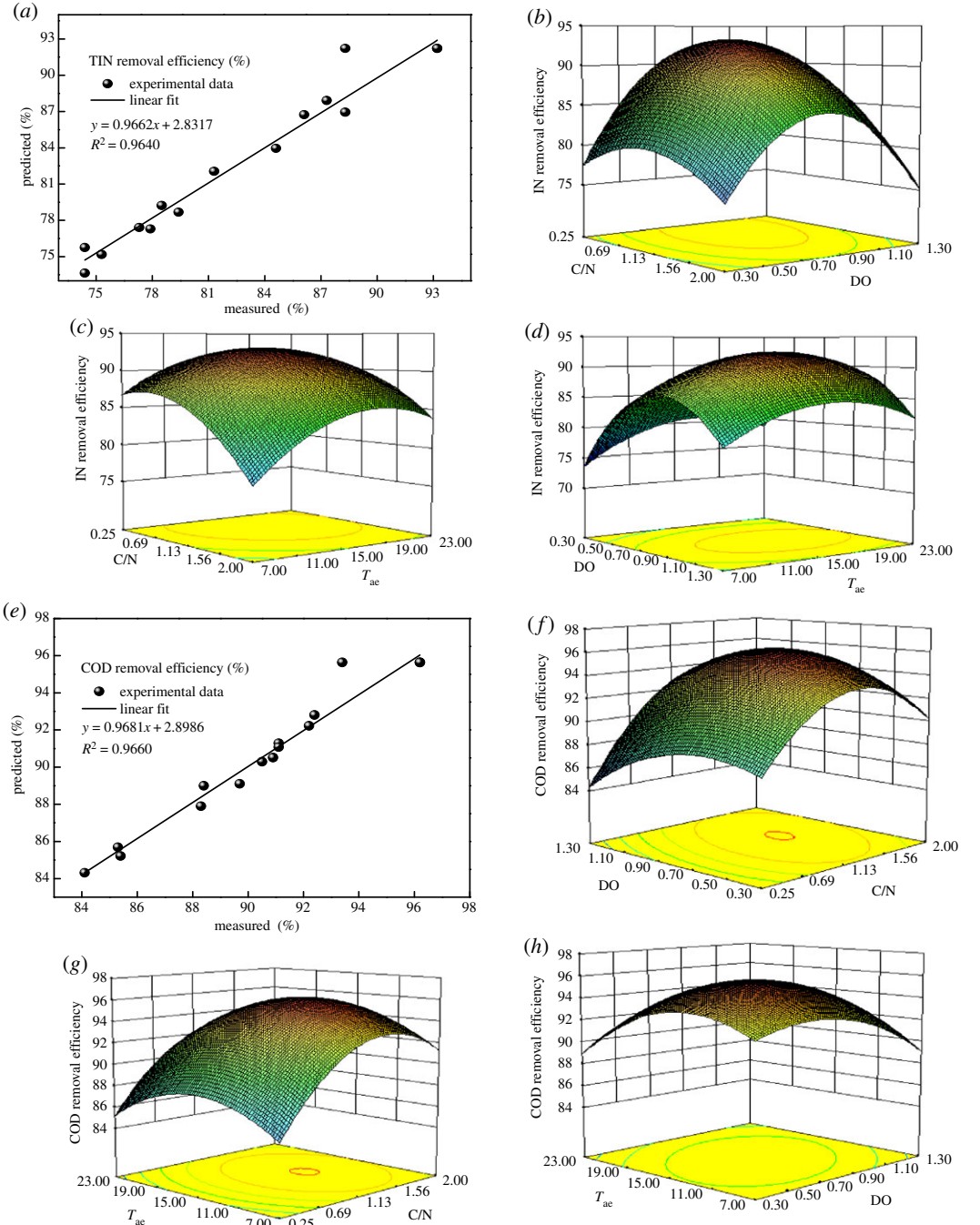

**Figure 2.** (*a*) Relationship between the measured and predicted TIN removal efficiency, (*b–d*) response 3D surface plot ($X_1$, $X_2$, $X_3$) for TIN removal efficiency at different C/N, DO and $T_{ae}$; (*e*) relationship between the measured and predicted COD removal efficiency, (*f–h*) response 3D surface plot ($X_1$, $X_2$, $X_3$) for COD removal efficiency at different C/N, DO and $T_{ae}$.

increases and then decreases with the increase in the C/N ratio. The $T_{ae}$ value has an insignificant effect on the removal efficiency of COD in the range of C/N ratio from 0.25 to 2, indicating that the C/N ratio has a greater effect on the COD removal efficiency than does $T_{ae}$. When the $T_{ae}$ value and C/N ratio are 12.95–13.93 min and 1.26–1.35, respectively, the COD removal efficiency is greater than 95.2%. The effect of the interaction between $T_{ae}$ and DO on the COD removal efficiency at a constant C/N ratio (1.13) is depicted in figure 2*h*. A concentric circle appears in the contour plot of the COD removal efficiency, indicating that both $T_{ae}$ and DO have an influence on COD removal efficiency, but the influence trend is not obvious. When $T_{ae}$ and DO are 12.62–15.98 min and 0.65–0.84 mg · l$^{-1}$, respectively, the removal efficiency of COD is greater than 95.5%.

According to the preceding experimental data, the desirability function of RSM software was used for analysis. The optimal conditions simulated by the software were as follows: the values of the C/N ratio,

DO and $T_{ae}$ were 1.16, 0.84 mg l$^{-1}$, and 15.75 min, respectively, and the corresponding simulated values of TIN and COD removal efficiencies were 92.31% and 95.67%, respectively (electronic supplementary material, figure S1). Subsequently, to verify the adequacy of the TIN and COD removal efficiencies predicted by the model, additional experiments were carried out according to the optimized parameters. After 37 days of stable operation, the results showed that the removal efficiencies of TIN and COD were, respectively, 93.4% and 91.6% (electronic supplementary material, figure S2). These results are basically consistent with those predicated by RSM, which verifies the validity of the model.

## 3.3. Microbial activity under different particle sizes

### 3.3.1. Analysis of sludge particle size distribution and sludge volume index (SVI$_{30}$)

Figure 3a,b presents the distributions of sludge particle size during the initial start-up and stable operation of the SNAD process, respectively. As shown in figure 3a, at the beginning of the SNAD process, the sludge particle size was mainly concentrated in the range of 0.2–0.6 mm, of which the particle size between 0.2 and 0.335 mm accounted for about 40% of the TSS, and the particle size between 0.335 and 0.6 mm accounted for about 25% of the TSS. The functional bacteria in anammox sludge can promote the adhesion between cells by secreting EPS, thus promoting the formation of anammox granular sludge [30]. During the stable operation of the SNAD process, the sludge particle size was mainly concentrated between 0.335 and 0.6 mm, accounting for about 46% of the TSS. The formation of SNAD granular sludge improves the settling performance of sludge and contributes to the stable operation of the SNAD system. As shown in figure 3c, during the start-up and optimization of the SNAD process, the SVI$_{30}$ values of Phase I, Phase II, Phase III and Phase IV were 62.1 ml g$^{-1}$· MLSS, 56.3 ml g$^{-1}$· MLSS, 53.3 ml g$^{-1}$· MLSS and 43.7 ml g$^{-1}$· MLSS, respectively. The SVI$_{30}$ value of Phase IV was 29.6% lower than that of Phase I, which indicates that the sedimentation performance of the activated sludge was improved.

### 3.3.2. Specific activity of bacteria under different particle sizes of sludge

In previous research on the SNAD system, there have been few studies on the effect of the specific activity of bacteria on nitrogen removal from different particle sizes of sludge [31–33]. To determine the specific activity of bacteria in different sludge particle sizes, three ranges of sizes were determined to represent SNAD granular sludge (less than 0.335 mm, 0.335–1.0 mm and greater than 1 mm). During denitrification, DNB$_a$ uses NO$_2^-$−N as an electron acceptor to improve nitrogen removal via heterotrophic denitrification (equation (1.4)). In addition, DNB$_b$ uses NO$_3^-$−N as an electron acceptor to remove nitrogen via heterotrophic denitrification (equation (1.3)).

As shown in figure 4, different PSDs affect the microbial community structure and microbial activity. Under the condition of a particle size of less than 0.335 mm, the specific activity of AerAOB was 4.73 mgN gVSS$^{-1}$ h$^{-1}$, which was 20.08% higher than that of AerAOB with a particle size of greater than 1 mm. The results indicate that the substances were more likely to be consumed by particles of a smaller size, and the specific activity of AerAOB tended to decrease with the increase in particle size. When the particle size was 0.335–1.0 mm, the specific activity of AnAOB was significantly improved (from 2.12 to 6.76 mgN gVSS$^{-1}$ h$^{-1}$), indicating that the larger granular sludge could provide the necessary anaerobic environment for AnAOB under the premise of ensuring mass transfer efficiency. Luo et al. [34] demonstrated that larger particles of granulated sludge harbour more microbial communities and more abundant AnAOB than small particles of granulated sludge. In addition, the specific activity of DNB$_a$ using NO$_3^-$−N as the reaction substrate, was higher than that of DNB$_b$ using NO$_2^-$−N as the reaction substrate. This confirms that with the increase in particle size, not only is the specific activity of AnAOB and DNB promoted, but the mass transfer efficiency of the substrate needed for AnAOB and DNB is not reduced.

## 3.4. Analysis of TMP and membrane fouling characteristics

During the operation of the reactor, the membrane flux was constant at about 20 l m$^{-2}$ h$^{-1}$. When the TMP increased to about 50 kPa, the membrane components were cleaned. The change of TMP in the SNAD system is presented in figure 5a. In Phase I (days 1–55), the TMP gradually increased from 8.2 to 51.1 kPa, and the MFR was 35.0 Pa h$^{-1}$. In Phase II (days 56–114), the TMP gradually increased from 9.4 to 51.8 kPa, and the MFR was 30.9 Pa h$^{-1}$. In Phase III (days 115–160), the TMP gradually

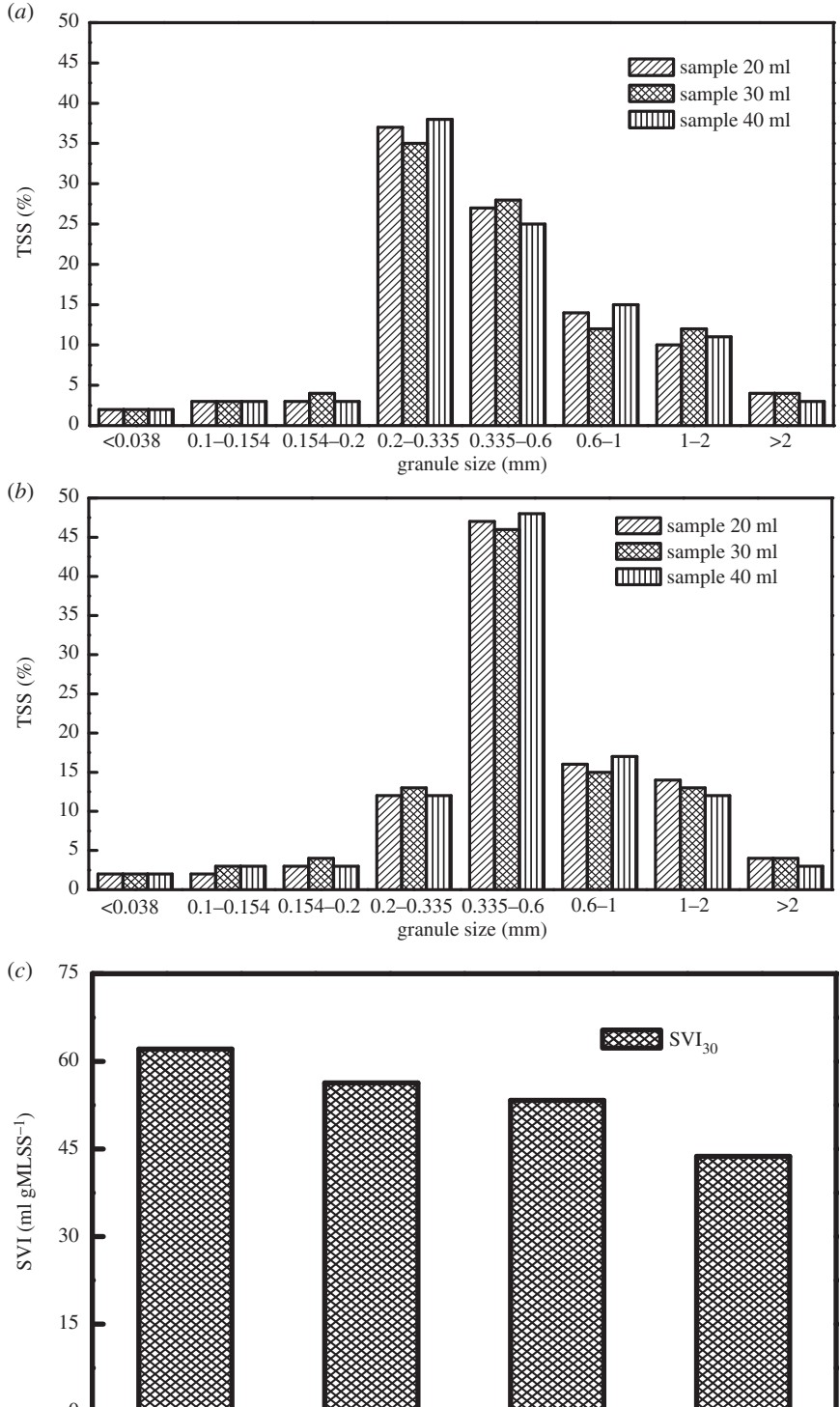

**Figure 3.** (a,b) The distributions of sludge particle size during the initial start-up and stable operation of the SNAD process; (c) SVI$_{30}$ at different stages in the SNAD system.

increased from 8.5 to 37.2 kPa, and the MFR was 25.9 Pa h$^{-1}$. In Phase IV (days 161–203), the TMP gradually increased from 8.7 to 29.3 kPa, and the MFR was 19.9 Pa h$^{-1}$. During the start-up and operation of the SNAD system, the TMP increased regularly. In Phase IV, the MFR decreased significantly compared with that in the previous three phases, which may be due to the slowing down of the MFR with the increase in sludge particle size. This observation indicates that the cultivation of granular sludge is an effective way to alleviate membrane fouling. The membrane filtration could keep almost all of the biomass within the reactor rather than being washed away with

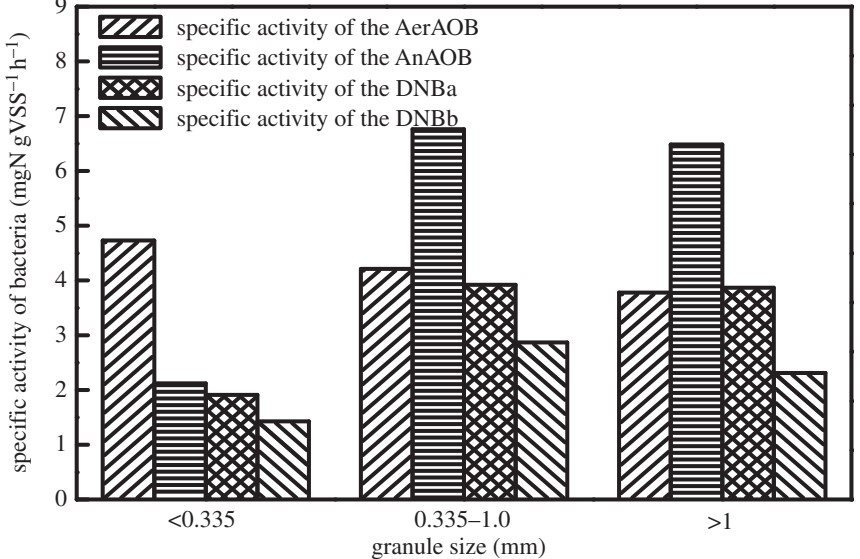

**Figure 4.** Microbial activity under different particle size.

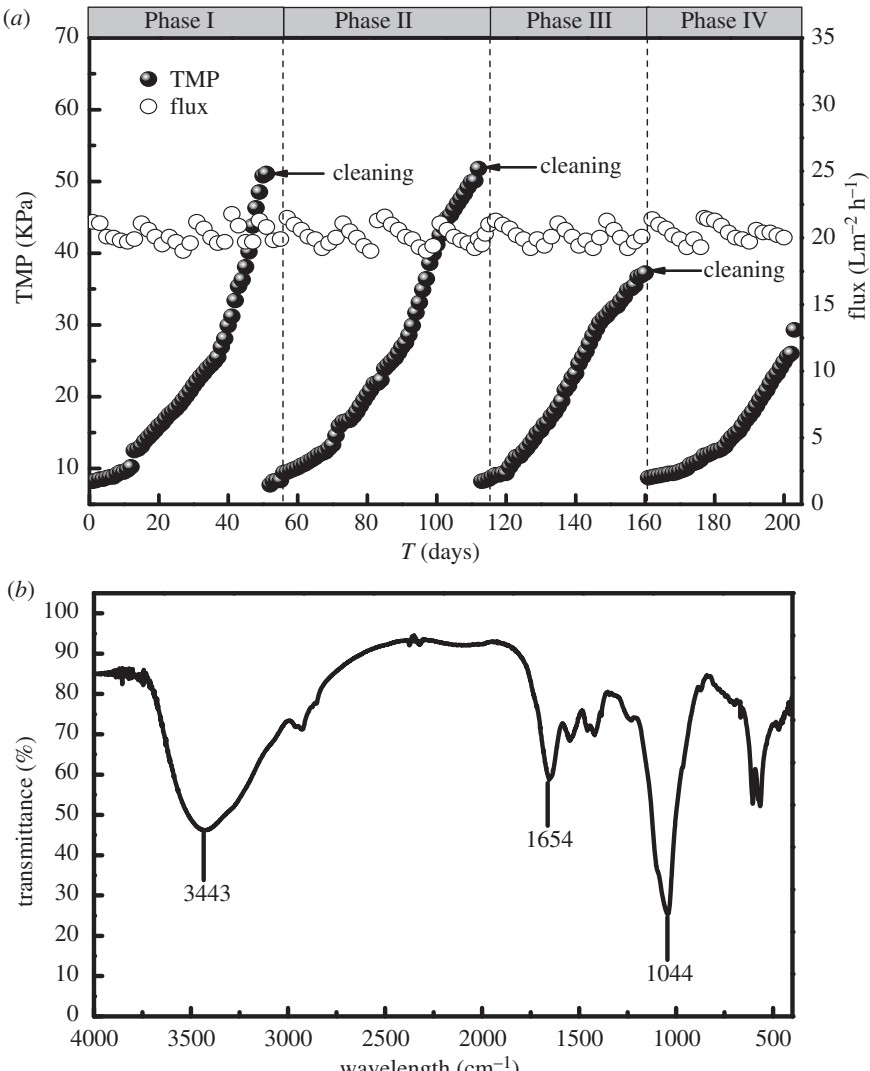

**Figure 5.** (*a*) TMP and membrane flux of the SNAD-MBR; (*b*) FT-IR spectra of the membrane foulants.

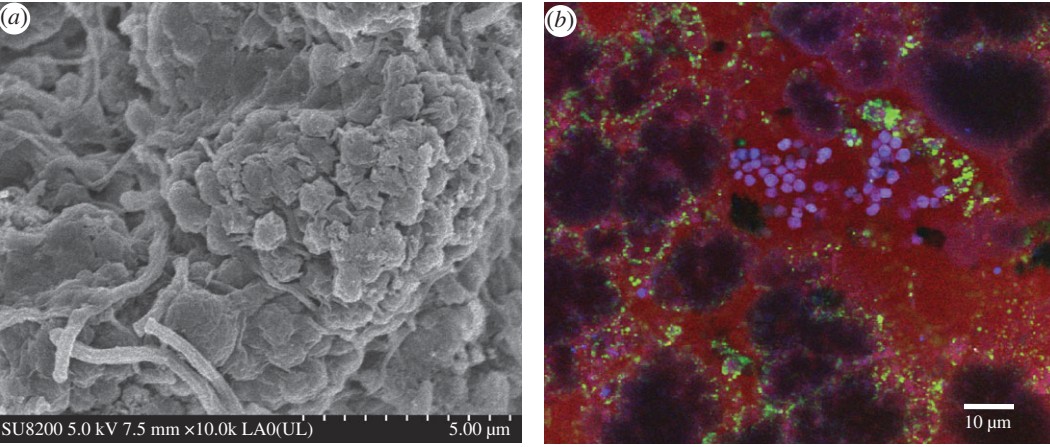

**Figure 6.** (*a*) SEM observation of the morphology and structure in biomass; (*b*) FISH images of sludge.

the effluent. MBR is more effective and suitable for cultivating slow-growing bacteria due to the complete separation of the SRT and HRT. During the operation of MBR, a large amount of soluble and colloidal substances were attached to the surface of the membrane to form a filter cake layer, which leads to the blockage of the membrane pores and further causes membrane fouling. Therefore, reducing the deposition of pollutants on the surface of membrane was the key to preventing membrane contamination. In this study, the deposition of pollutants on the membrane surface was effectively controlled by the shear force generated by the intermittent aeration with the assistance of the magnetic agitator.

In addition, Fourier transform infrared spectroscopy (FT-IR) was employed to analyse the fouling substances on the surface of the membrane. As shown in figure 5*b*, the absorption peak occurred at 3433 cm$^{-1}$, and was mainly generated by the N–H stretching vibration of amide. The absorption peak at 1654 cm$^{-1}$ mainly characterizes amides in the secondary structures of proteins. In addition, the absorption peak at 1044 cm$^{-1}$ is attributable to the functional groups of polysaccharides. The results of the present study reveal that the pollutants on the membrane surface are mainly composed of proteins and polysaccharides.

## 3.5. Analysis of microbial community in the SNAD system

The coexistence and community structure of the AerAOB, AnAOB and DNB in the SNAD process were confirmed by SEM and FISH. As presented in figure 6*a*, the bacteria morphologies were mainly spherical-shaped, short rod-shaped and filamentous. Specifically, the AnAOB mainly existed in the form of irregular spherical-shaped and cauliflower-shaped aggregates, and the main types of AerAOB were spherical-shaped. Finally, DNB mainly existed in the form of filamentous bacteria. The spatial distribution of AerAOB, AnAOB and DNB in the SNAD sludge was further analysed by the FISH method. As shown in figure 6*b*, the FISH analysis further proves the coexistence of AerAOB, AnAOB and DNB in the SNAD system. The proportion of AerAOB (blue) to the area of the visible region represents the relative value of the AerAOB quantity, the proportion of AnAOB (red) to the area of visible region represents the relative value of the AnAOB quantity and the proportion of DNB (green) to the area of the visible region represents the relative value of the DNB quantity. AerAOB, AnAOB and DNB accounted for 31.3%, 39.5% and 29.8% of the total bacteria, respectively. While the AerAOB mainly appeared in the aerobic area on the surface of the granulated sludge, the AnAOB and DNB appeared in the anoxic area inside the granulated sludge.

## 4. Conclusion

In this study, an MBR reactor was successfully used to realize the SNAD process of simultaneous nitrogen and carbon removal. The results reveal that the optimal conditions were a C/N ratio of 1.16, a DO concentration of 0.84 mg l$^{-1}$ and a $T_{ae}$ value of 15.75 min. Under these conditions, the SNAD process achieved the effect of simultaneous nitrogen and carbon removal, and the removal efficiencies of TIN and COD were 92.31% and 95.67%, respectively.

Data accessibility. Data available from the Dryad Digital Repository: https://doi.org/10.5061/dryad.tht76hdwg.

Authors' contributions. K.Z. and Z.W. conceived of the study, participated in the design of the study and drafted the manuscript. M.S., D.L and L.H. collected field data. J.Z., X.W. and J.L. coordinated the study. All authors gave final approval for publication.

Competing interests. The authors declare no competing interests.

Funding. This study was funded by Major Science and Technology Program for Water Pollution Control and Treatment (grant no. 2018ZX07701001-25).

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
