## [Reviewer comments · Royal Society Open Science]

Review History

RSOS-200584.R0 (Original submission)

Review form: Reviewer 1

Is the manuscript scientifically sound in its present form?

Yes

Are the interpretations and conclusions justified by the results?

Yes

Is the language acceptable?

Yes

Do you have any ethical concerns with this paper?

No

Have you any concerns about statistical analyses in this paper?

No

Recommendation?

Accept as is

Comments to the Author(s)

Through response surface method, the successful operation of SNAD in MBR was realized, with both high removal of TIN and COD, and reduced membrane fouling. The optimal condition of TN/COD DO T_{ae}, was proved.

Review form: Reviewer 2**Is the manuscript scientifically sound in its present form?**

No

Are the interpretations and conclusions justified by the results?

Yes

Is the language acceptable?

Yes

Do you have any ethical concerns with this paper?

No

Have you any concerns about statistical analyses in this paper?

Yes

Recommendation?

Major revision is needed (please make suggestions in comments)

Comments to the Author(s)

In this paper, the Authors aimed at assessing optimization of operational parameters of the SNAD process in the membrane reactor. This is a process of both scientific and practical interest. The Authors operate the system for more than 200 days with robust monitoring analyzes that allow to achieve the objectives. In addition, there is an analysis of microbial activity under different particle sizes and FISH analyzes.

Although, the results presented are difficult to follow and interpret. The results discussed in section 3.1 are very confusing. I suggest a table like table 1, which each phase could be divided into three lines, one for each condition tested, and add the efficiency for removing COD and TIN and ammonia with average values, number of samples and standard deviation. Less effort should be made in describing changes in concentrations but the main changes of the operational parameter on the process.

It is not clear how the experimental results are exhibited in Table 2S were obtained. There are some conditions where the TIN removal efficiency was less than 70% while the graph in figure 2 the measured efficiencies start at 75%.

Information should be added about the periodicity of dissolved oxygen measurements, and the variability of monitoring, since it is difficult to maintain constant low oxygen concentrations with aeration systems.

Taking the above comments into account, I recommend that the paper could further be considered for publication after major revisions.

Specify comments:

1. (Page 3, line 6) The aeration time parameter is not well defined in the summary, since the use of intermittent aeration is not mentioned.
2. (Page 4, lines 14-16) The following statement is unclear: "the simultaneous removal ... in a single reactor is realized by intermittent aeration".
3. (Page 8, line 6) Specify methods used for nitrite and nitrate ammonia.
4. (Page 9, line 22) FT-IR, define the acronym.
5. (page 9, line 8) The statement "...was then injected into 500-mL beakers to test the microbial activity" is not clear, what is the volume of the media added?
6. (Page 10, line 6) SEM define the acronym.
7. (Page 10, line 13) FISH define the acronym.
8. (Page 11, lines 4-5) The statement "The start-up and 5 optimizations of the SNAD process were operated for 203 days" is not very accurate. The reactor was operated for 203 days.
9. (Page 11, line 5) Remove "During the operation of the reactor"
10. (Page 11, line 12) COD could also inhibit the AOB.
11. (Page 12, line 3) The statement "... and the intermittent aeration mode (aeration time to non-aeration time (min:min)) and the DO were maintained at 7:4.." is in conflict with the Table 1, where indicates the aeration time was reduced to 7:7. Also standardize what comes first, the time of aeration or non-aeration.
12. (Page 13 lines 8-9) The statement "The non-aeration time was maintained at 4 min..." is in conflict with the Table 1 that present the non-aeration period of 7 min
13. (Page 14, line 6) In the statement "In the last stage (from day 188 to day 203)" what is the difference between the last stage and the previous one ? if the OD was 1.3 indicate in the text and correct Table 1, which shows that the OD varied from 0.3 to 0.8. In this last phase there is only a description of the results, an analysis of the impact of OD on the processes involved in the reactor is missing, especially since the last step (OD equal to 1.3?) seems to have been better because it has greater ammonia removal.
14. Indicate all operation changes in Fig 1, not only the phases.
15. Scale bar is missing in Fig 6 (b) and a description of what the colors represent should be inserted in caption for clarity.

Review form: Reviewer 3

Is the manuscript scientifically sound in its present form?

Yes

Are the interpretations and conclusions justified by the results?

Yes

Is the language acceptable?

Yes

Do you have any ethical concerns with this paper?

No

Have you any concerns about statistical analyses in this paper?

No

Recommendation?

Major revision is needed (please make suggestions in comments)

Comments to the Author(s)

In this study, an MBR reactor was successfully used to realize the SNAD process of simultaneous nitrogen and carbon removal, and the optimal conditions were obtained. The SNAD process achieved the effect of simultaneous nitrogen and carbon removal, and the removal efficiencies of TIN and COD were 92.31% and 95.67%, respectively at C/N ratio of 1.16, a DO concentration of 0.84 mg/L, and a T_{ae} value of 15.75 min. The work is sound and the contents are informative. In my opinion, a major revision is needed after addressing the following aspects.

--SNAD process has recently been widely researched. Please clarify the key points to realize the process with the results.

--Please further specify the research difficulties and features of the present research in discussion section.

--What is the relationship between the phasing time and TMP. It is a little one-sided to confirm the composition only from the FTIR analysis.

--The references should be updated with recent publications on this topic, such as *Bioresource Technol* 2019, 283: 138-147; *Water Res* 2020, 174: 115632, etc.

Decision letter (RSOS-200584.R0)

Dear Professor Jun:

Title: Optimization of nitrogen and carbon removal with simultaneous partial nitrification, anammox, and denitrification (SNAD) in membrane bioreactor (MBR)

Manuscript ID: RSOS-200584

The editor assigned to your manuscript has now received comments from reviewers. We would like you to revise your paper in accordance with the referee and Subject Editor suggestions which can be found below (not including confidential reports to the Editor). Please note this decision does not guarantee eventual acceptance.

Please submit your revised paper before 19-Jun-2020. Please note that the revision deadline will expire at 00.00am on this date. If we do not hear from you within this time then it will be assumed that the paper has been withdrawn. In exceptional circumstances, extensions may be possible if agreed with the Editorial Office in advance. We do not allow multiple rounds of

revision so we urge you to make every effort to fully address all of the comments at this stage. If deemed necessary by the Editors, your manuscript will be sent back to one or more of the original reviewers for assessment. If the original reviewers are not available we may invite new reviewers.

RSC Associate Editor:
Comments to the Author:
(There are no comments.)

RSC Subject Editor:
Comments to the Author:
(There are no comments.)

Reviewers' Comments to Author:
Reviewer: 1

Comments to the Author(s)
through response surface method, the successful operation of SNAD in MBR was realized, with both high removal of TIN and COD, and reduced membrane fouling .the optimal condition of TN/COD DO Tae, was proved.

Reviewer: 2

Comments to the Author(s)

In this paper, the Authors aimed at assessing optimization of operational parameters of the SNAD process in the membrane reactor. This is a process is both scientific and practical interest. The Authors operate or the system for more than 200 days with robust monitoring analyzes that allow to achieve the objectives. In addition, there is an analysis of microbial activity under different particle sizes and FISH analyzes.

Although, the results presented are difficult to follow and interpret. The results discussed in section 3.1 are very confusing. I suggest a table like table 1, which each phase could be divided into three lines, one for each condition tested, and add the efficiency for removing COD and TIN and ammonia with average values, number of samples and standard deviation. Less effort should be made in describing changes in concentrations but the main changes of the operational parameter on the process.

Is not clear how the experimental results are exhibited in Table 2S were obtained. There are some conditions where the TIN removal efficiency was less than 70% while the graph in figure 2 the measured efficiencies start at 75%.

Information should be added about the periodicity of dissolved oxygen measurements, and the variability of monitoring, since it is difficult to maintain constant low oxygen concentrations with aeration systems.

Taking the above comments into account, I recommend that the paper could further be considered for publication after major revisions.

Specify comments:

1. (Page 3, line 6) The aeration time parameter is not well defined in the summary, since the use of intermittent aeration is not mentioned.
2. (Page 4, lines 14-16) The following statement is unclear: "the simultaneous removal ... in a single reactor is realized by intermittent aeration".
3. (Page 8, line 6) Specify methods used for nitrite and nitrate ammonia.
4. (Page 9, line 22) FT-IR, define the acronym.
5. (page 9, line 8) The statement "...was then injected into 500-mL beakers to test the microbial activity" is not clear, what is the volume of the media added?
6. (Page 10, line 6) SEM define the acronym.
7. (Page 10, line 13) FISH define the acronym.
8. (Page 11, lines 4-5) The statement "The start-up and 5 optimizations of the SNAD process were operated for 203 days" is not very accurate. The reactor was operated for 203 days.
9. (Page 11, line 5) Remove "During the operation of the reactor"
10. (Page 11, line 12) COD could also inhibit the AOB.
11. (Page 12, line 3) The statement "... and the intermittent aeration mode (aeration time to non-aeration time (min:min)) and the DO were maintained at 7:4.." is in conflict with the Table 1,

where indicates the aeration time was reduced to 7:7. Also standardize what comes first, the time of aeration or non-aeration.

12. (Page 13 lines 8-9) The statement “The non-aeration time was maintained at 4 min...” is in conflict with the Table 1 that present the non-aeration period of 7 min

13. (Page 14, line 6) In the statement “In the last stage (from day 188 to day 203)” what is the difference between the last stage and the previous one ? if the OD was 1.3 indicate in the text and correct Table 1, which shows that the OD varied from 0.3 to 0.8. In this last phase there is only a description of the results, an analysis of the impact of OD on the processes involved in the reactor is missing, especially since the last step (OD equal to 1.3?) seems to have been better because it has greater ammonia removal.

14. Indicate all operation changes in Fig 1, not only the phases.

15. Scale bar is missing in Fig 6 (b) and a description of what the colors represent should be inserted in caption for clarity.

Reviewer: 3

Comments to the Author(s)

In this study, an MBR reactor was successfully used to realize the SNAD process of simultaneous nitrogen and carbon removal, and the optimal conditions were obtained. The SNAD process achieved the effect of simultaneous nitrogen and carbon removal, and the removal efficiencies of TIN and COD were 92.31% and 95.67%, respectively at C/N ratio of 1.16, a DO concentration of 0.84 mg/L, and a Tae value of 15.75 min. The work is sound and the contents are informative. In my opinion, a major revision is needed after addressing the following aspects.

--SNAD process has recently been widely researched. Please clarify the key points to realize the process with the results.

--Please further specify the research difficulties and features of the present research in discussion section.

--What is the relationship between the phasing time and TMP. It is a little one-sided to confirm the composition only from the FTIR analysis.

--The references should be updated with recent publications on this topic, such as Bioresource Technol 2019, 283: 138-147; Water Res 2020, 174: 115632, etc.

Author's Response to Decision Letter for (RSOS-200584.R0)

See Appendix A.

RSOS-200584.R1 (Revision)

Review form: Reviewer 2

Is the manuscript scientifically sound in its present form?

No

Are the interpretations and conclusions justified by the results?

Yes

Is the language acceptable?

Yes

Do you have any ethical concerns with this paper?

No

Have you any concerns about statistical analyses in this paper?

Yes

Recommendation?

Major revision is needed (please make suggestions in comments)

Comments to the Author(s)

The vast majority of previous comments have been satisfactorily clarified and resolved. However, I still have some key concerns that need to be clarified.

Still not clear to me how the experimental results exhibited in Table 2S were obtained. During the operational period the conditions tested are not the same as in table 2S. How were these experiments done? When DO concentration of 1.3 were tested? How accurate was it to keep DO levels at 0.3, 0.8 and 1.3?

All operational changes are still being difficult to follow. If it is difficult to indicate them in the figure, you can try to indicate in table 1 with more rows indicating what the conditions were and their respective period of days.

There are some conditions where the TIN removal efficiency was less than 70% while the graph in figure 2 the measured efficiencies start at 75%.

Information should be added about the periodicity of dissolved oxygen measurements, and the variability of monitoring, since it is difficult to maintain constant low oxygen concentrations with aeration systems and this was an important variable in the process.

The authors present sufficient results and of high scientific interest, although without these clarifications, in the present form of the manuscript, I do not recommend it for publication.

Review form: Reviewer 3

Is the manuscript scientifically sound in its present form?

Yes

Are the interpretations and conclusions justified by the results?

Yes

Is the language acceptable?

Yes

Do you have any ethical concerns with this paper?

No

Have you any concerns about statistical analyses in this paper?

No

Recommendation?

Accept with minor revision (please list in comments)

Comments to the Author(s)

The authors have made some modifications and the responses are appreciated. The only reminder is that some recent publication on this topic should be updated.

Decision letter (RSOS-200584.R1)

Dear Professor Jun:

Title: Optimization of nitrogen and carbon removal with simultaneous partial nitrification, anammox, and denitrification (SNAD) in membrane bioreactor (MBR)
Manuscript ID: RSOS-200584.R1

The editor assigned to your paper has now received comments from reviewers. We would like you to revise your paper in accordance with the referee and Subject Editor suggestions which can be found below (not including confidential reports to the Editor). Please note this decision does not guarantee eventual acceptance.

Please submit a copy of your revised paper before 02-Aug-2020. Please note that the revision deadline will expire at 00.00am on this date. If we do not hear from you within this time then it will be assumed that the paper has been withdrawn. In exceptional circumstances, extensions may be possible if agreed with the Editorial Office in advance. We do not allow multiple rounds of revision so we urge you to make every effort to fully address all of the comments at this stage. If deemed necessary by the Editors, your manuscript will be sent back to one or more of the original reviewers for assessment. If the original reviewers are not available we may invite new reviewers.

Yours sincerely,
Dr Laura Smith

Publishing Editor, Journals

RSC Associate Editor:
Comments to the Author:
(There are no comments.)

RSC Subject Editor:
Comments to the Author:
(There are no comments.)

Reviewers' Comments to Author:
Reviewer: 2

Comments to the Author(s)
The vast majority of previous comments have been satisfactorily clarified and resolved. However, I still have some key concerns that need to be clarified.

Still not clear to me how the experimental results exhibited in Table 2S were obtained. During the operational period the conditions tested are not the same as in table 2S. How were these experiments done? When DO concentration of 1.3 were tested? How accurate was it to keep DO levels at 0.3, 0.8 and 1.3?

All operational changes are still being difficult to follow. If it is difficult to indicate them in the figure, you can try to indicate in table 1 with more rows indicating what the conditions were and their respective period of days.

There are some conditions where the TIN removal efficiency was less than 70% while the graph in figure 2 the measured efficiencies start at 75%.

Information should be added about the periodicity of dissolved oxygen measurements, and the variability of monitoring, since it is difficult to maintain constant low oxygen concentrations with aeration systems and this was an important variable in the process.

The authors present sufficient results and of high scientific interest, although without these clarifications, in the present form of the manuscript, I do not recommend it for publication.

Reviewer: 3

Comments to the Author(s)
The authors have made some modifications and the responses are appreciated. The only reminder is that some recent publication on this topic should be updated.

Author's Response to Decision Letter for (RSOS-200584.R1)

See Appendix B.

Decision letter (RSOS-200584.R2)

Dear Professor Jun:

Title: Optimization of nitrogen and carbon removal with simultaneous partial nitrification, anammox, and denitrification (SNAD) in membrane bioreactor (MBR)
Manuscript ID: RSOS-200584.R2

It is a pleasure to accept your manuscript in its current form for publication in Royal Society Open Science. The chemistry content of Royal Society Open Science is published in collaboration with the Royal Society of Chemistry.

RSC Associate Editor
Comments to the Author:
(There are no comments.)

Reviewer(s)' Comments to Author:

Appendix A

Dear Dr Laura Smith,

Thank you so much for giving us the opportunity to revise and resubmit our manuscript (RSOS-200584), entitled ‘ Optimization of nitrogen and carbon removal with simultaneous partial nitrification, anammox, and denitrification (SNAD) in membrane bioreactor (MBR)’. We sincerely thank you and the reviewer for your valuable feedback that we have used to improve the quality of our manuscript. The reviewer comments are laid out below in italicized font and specific concerns have been numbered. Our response is given in normal font and changes/additions to the manuscript are given in blue text.

We hope that the revised version of the manuscript could be considered for publication in your journal. I look forward to hearing from you soon.

With best wishes,

Yours sincerely,

Kai Zhang,

Jun 18, 2020

To reviewer:

We sincerely thank you for your professional review work on our manuscript. Your thoughtful comments and constructive suggestions have contributed a lot to improve the quality of our manuscript. As you are concerned, there are several problems that need to be addressed. According to your nice suggestions, we have made extensive corrections to our previous draft. After this revision, we have written a point-by-point response letter to you as you can see above. And the detailed corrections are listed below.

Reviewer 1

Comments to the Author(s)

Through response surface method, the successful operation of SNAD in MBR was realized, with

both high removal of TIN and COD, and reduced membrane fouling .the optimal condition of TN/COD DO Tae, was proved.

Response: Thanks to the recognition of the reviewer, we will continue to make efforts in the future research.

Reviewer 2

Comment 1: *(Page 3, line 6) The aeration time parameter is not well defined in the summary, since the use of intermittent aeration is not mentioned.*

Response: Thank you so much for your valuable comments about our abstract part. According to your thoughtful suggestion, we have defined and supplemented the intermittent aeration mode. The additions were ‘During the entire experiment, the intermittent aeration(non-aerobic time: aeration time, min/min) cycle was controlled by a time-controlled switch, and the aeration rate was controlled by a gas flowmeter,and the optimal operating parameters as determined by response surface methodology (RSM) were a C/N value of 1.16, a DO value of 0.84 mg·L⁻¹, and an aerobic time (Tae) of 15.75 min’.In the revised manuscript we have marked these changes in blue.. We hope our revised manuscript can be up to your standard.

Comment 2: *(Page 4, lines 14-16) The following statement is unclear: “the simultaneous removal ... in a single reactor is realized by intermittent aeration”.*

Response: Thank you for your careful check. We have corrected ‘Therefore, using intermittent aeration mode, SNAD can simultaneously remove nitrogen and carbon in a single reactor.’ In the revised manuscript we have marked these changes in blue. We feel great thanks for you to point out our mistake.

Comment 3: *(Page 8, line 6) Specify methods used for nitrite and nitrate ammonia.*

Response: We sincerely thank you for your professional review work We use spectrophotometry to determine the concentration of nitrite and nitrate. The absorbance values of a series of standard concentrations were determined by using solutions of nitrite nitrogen standard and nitrate nitrogen standard respectively,and then the standard curves of the two were drawn respectively. Finally, the absorbance values measured by spectrophotometer were substituted into the standard curves to obtain the corresponding concentration values.We hope our response can be up to your standard.

Comment 4: *(Page 9, line 22) FT-IR, define the acronym.*

Response: Thank you for your careful check. According to your suggestion, we have defined the full spelling of FT-IR. The additions were ‘The Fourier-Transform infrared spectroscopy (FT-IR) analysis.’ In the revised manuscript we have marked these changes in blue.

Comment 5: (page 9, line 8) *The statement “...was then injected into 500-mL beakers to test the microbial activity” is not clear, what is the volume of the media added?*

Response: We sincerely thank you for your professional review work on our manuscript. We have corrected ‘Before the batch test, the activated sludge was washed three times, and 15 g of wet sludge was then injected into 500-mL beakers to test the microbial activity. Finally, the total volume of adding 15 g wet sludge and medium was 500 ml.’ The batch experiment conditions are described in Table 1S. In the revised manuscript we have marked these changes in blue.

Comment 6: (Page 10, line 6) *SEM define the acronym.*

Response: Thank you for your careful check. According to your suggestion, we have defined the full spelling of SEM. The additions were ‘Scanning Electron Microscope (SEM) observation’ In the revised manuscript we have marked these changes in blue.

Comment 7: (Page 10, line 13) *FISH define the acronym.*

Response: Thank you for your careful check. According to your suggestion, we have defined the full spelling of FISH. The additions were ‘Fluorescence In Situ Hybrid (FISH) analysis’ In the revised manuscript we have marked these changes in blue.

Comment 8: (Page 11, lines 4-5) *The statement “The start-up and 5 optimizations of the SNAD process were operated for 203 days” is not very accurate. The reactor was operated for 203 days.*

Response: Thank you for your careful check. According to your suggestion, we have corrected ‘The reactor was operated for 203 days’. In the revised manuscript we have marked these changes in blue.

Comment 9: (Page 11, line 5) Remove *"During the operation of the reactor"*

Response: According to your suggestion, we have removed "During the operation of the Reactor".

Comment 10: (Page 11, line 12) COD could also inhibit the AOB.

Response: We sincerely thank you for your professional review work on our manuscript. Through the phenomenon of this experiment and the summary of published literature, we found that the influence of organic matter on nitrification was mainly manifested as the competition between heterotrophic bacteria and autotrophic bacteria for DO. When temperature and pH are suitable, DO and ammonia nitrogen supply is sufficient, and the concentration of organic matter does not affect nitrification. When DO was insufficient and organic matter concentration was high, the competition of heterotrophic bacteria for DO in water was stronger than that of nitrifying bacteria, and the growth of nitrifying bacteria was inhibited. The stability of short-cut nitrification is affected.¹⁻² In this experiment, intermittent aeration and sufficient influent $\text{NH}_4^+\text{-N}$ concentration provided sufficient substrate and reaction conditions for AOB, so the addition of organic matter did not completely inhibit nitrification. I will continue to study in future studies to verify them.

[1] Jang Y., Guan-hong T. (2007). Short-cut nitrification in domestic wastewater treatment by using membrane bioactor . Environmental Science and Technology, 30(11):95-97.

[2] Li, Q., Li, P., Zhu, P., Wu, J., & Liang, S. (2008). Effects of exogenous organic carbon substrates on nitrous oxide emissions during the denitrification process of sequence batch reactors. Environmental Engineering Science, 25, 1221–1228.

Comment 11: (Page 12, line 3) The statement *"... and the intermittent aeration mode (aeration time to non-aeration time (min:min)) and the DO were maintained at 7:4.."* is in conflict with the Table 1, where indicates the aeration time was reduced to 7:7. Also standardize what comes first, the time of aeration or non-aeration.

Response: Thank you for your careful check. We feel sorry about our mistakes and carelessness in the content. According to your suggestion, we have rechecked the data in the manuscript and Table 1, and standardized the non-aeration and aeration time in the manuscript. In the revised manuscript we have marked these changes in blue. We feel great thanks for you to point out our mistakes.

Comment 12: (Page 13 lines 8-9) The statement *"The non-aeration time was maintained at 4*

min...” is in conflict with the Table 1 that present the non-aeration period of 7 min

Response: Thank you for your careful check. Comment 12 and 11 fall into the same category of errors. We have checked the manuscript. We are sorry again for our mistake. Thank you very much for pointing out our mistake. In the revised manuscript we have marked these changes in blue.

Comment 13: (Page 14, line 6) *In the statement “In the last stage (from day 188 to day 203)” what is the difference between the last stage and the previous one ? if the OD was 1.3 indicate in the text and correct Table 1, which shows that the OD varied from 0.3 to 0.8. In this last phase there is only a description of the results, an analysis of the impact of OD on the processes involved in the reactor is missing, especially since the last step (OD equal to 1.3?) seems to have been better because it has greater ammonia removal.*

Response: We sincerely thank you for your professional review work on our manuscript. In Phase IV, the intermittent aeration mode (non-aeration time to aeration time (min:min) ratio) was maintained at 7:15. From day 172 to day 203, the DO value in the aeration stage was 0.8 mg/L, which was adjusted by increasing the aeration rate. Therefore, in the three stages of the IV phase, the aeration rate of the first process (from day 161 to day 171) is 500 mL/min, the second process (from day 172 to day 187) and the third process (from day 188 to day 203) is 700 mL/min. When we analyzed the data, we found that the data fluctuated greatly from day 172 to day 187 and stabilized from day 188 to day 203, so we divided the data from day 172 to day 203 into two processes for explanation. In this experiment, only the influence of the change of DO value in the reactor on functional bacteria was studied, so the relationship between the aeration rate and DO value was not systematically analyzed.

According to your suggestion, we have added an analysis of the impact of DO on different processes in the reactor. The additions were ‘During the period in which the DO was $0.8 \text{ mg}\cdot\text{L}^{-1}$ (from day 172 to day 203). With the increase of DO, $\text{NH}_4^+\text{-N}$ removal rate tends to increase, indicating that DO not only meets the requirements of heterotrophic bacteria, but also provides sufficient DO environment for AOB. In addition, with the decrease of $\text{NH}_4^+\text{-N}$ concentration and TIN concentration of effluent, anammox may occur in the reactor.’ In the revised manuscript we have marked these changes in blue.

Comment 14: *Indicate all operation changes in Fig 1, not only the phases.*

Response: Thank you so much for your advice. After adding all the operation changes in Fig 1, it will appear very messy and cannot be explained clearly. Therefore, other operation conditions are explained in Table 1. According to your thoughtful suggestion, we have supplemented the aeration rate data in Table 1. We've marked the changes and additions in blue. We hope our revised

manuscript can be up to your standard.

Comment 15: *Scale bar is missing in Fig 6 (b) and a description of what the colors represent should be inserted in caption for clarity.*

Response: Thank you for your careful check. According to your thoughtful suggestion, we have added the scale bar in Fig 6(b) and inserted a description of what the color represented in caption. In the revised manuscript we have marked these changes in blue.

Reviewer 3

Comment 1: *SNAD process has recently been widely researched. Please clarify the key points to realize the process with the results.*

Response: We sincerely thank you for your professional review work. In this study, an MBR reactor was successfully used to realize the SNAD process of simultaneous nitrogen and carbon removal. During the entire experiment, the intermittent aeration (non-aerobic time: aeration time, min/min) cycle was controlled by a time-controlled switch. In the MBR, activated sludge was firstly inoculated for domesticating and enriching AOB to achieve stable short-cut nitrification, and then anaerobic ammonification flocculent sludge was added to achieve stable operation of the CANON process. Finally, the desirability function of RSM software was used for analysis. The optimal conditions simulated by the software were as follows: the values of the C/N ratio, DO, and T_{ae} were 1.16, 0.84 mg·L⁻¹, and 15.75 min, respectively, and the corresponding simulated values of TIN and COD removal efficiencies were 92.31% and 95.67%, respectively. Subsequently, to verify the adequacy of the TIN and COD removal efficiencies predicted by the model, additional experiments were carried out according to the optimized parameters. After 37 days of stable operation, the results showed that the removal efficiencies of TIN and COD were respectively 93.4% and 91.6%. We hope our response and revised manuscript can be up to your standard.

Comment 2: *Please further specify the research difficulties and features of the present research in discussion section.*

Response: Thank you so much for your valuable comments about our discussion section. According to your thoughtful suggestion, we have added the difficulties and features of this study in the discussion section. The additions were 'In summary, the SNAD process operated stably in the intermittent aeration MBR system. In this study, three main biochemical reactions are involved in the SNAD process. Under the condition of intermittent aeration, partial ammonium is oxidized to nitrite by AOB, and then the remaining NH₄⁺-N and the formed NO₂⁻-N act as the substrate of AnAOB to form NO₃⁻-N and N₂. Finally, the residue of NO₂⁻-N and the produced NO₃⁻-N are reduced

to N₂ through denitrifying process. Accordingly, how to optimize the intermittent aeration and C/N ratio becomes the difficulty of SNAD process stable operation.’. ‘The membrane filtration could keep almost all of the biomass within the reactor rather than being washed away with the effluent. MBR is more effective and suitable for cultivating slow-growing bacteria due to the complete separation of the SRT and HRT. During the operation of MBR, a large amount of soluble and colloidal substances were attached to the surface of the membrane to form a filter cake layer, which leads to the blockage of the membrane pores and further causes membrane fouling. Therefore, reducing the deposition of pollutants on the surface of membrane was the key to prevent membrane contamination. In this study, the deposition of pollutants on the membrane surface was effectively controlled by the shear force generated by the intermittent aeration with the assisted of the magnetic agitator.’ We've marked the changes and additions in blue. We hope our revised manuscript can be up to your standard.

Comment3: *What is the relationship between the phasing time and TMP. It is a little one-sided to confirm the composition only from the FTIR analysis.*

Response: We sincerely thank you for your professional review work on our manuscript. There is no necessary relationship between the four phases and TMP. In this study, membrane components were chemically cleaned when TMP reached 50kPa. The changes of sludge particle size at different phases and the characteristics of the specific activity of bacteria under different particle sizes were investigated. Therefore, in phase III and IV, the membrane components were not cleaned when the TMP reached 50kPa. Instead, the membrane components were cleaned at the end of each phase and the MFR was used to characterize the membrane component pollution. The experiment has already been finished and the sludge sample was not preserved. Due to the Novel Coronavirus effect, our school laboratory has been closed. We feel very sorry that we cannot provide sufficient support for other data on the composition analysis of membrane pollutants. In the future study, We will investigate the composition analysis of membrane pollutants according to your suggestions. Thank you again for your valuable advice. We deeply apologize for your inconvenience. If there is any need, please don't hesitate to contact me.

Comment4: *The references should be updated with recent publications on this topic, such as Bioresource Technol 2019, 283: 138-147; Water Res 2020, 174: 115632, etc.*

Response: We sincerely thank you for your thoughtful suggestion on our manuscript. According to your suggestion, we have reviewed the latest literature and updated the corresponding references in the manuscript. In the revised manuscript we have marked these changes in blue.

Appendix B

Dear Dr Laura Smith,

Thank you so much for giving us the opportunity to revise and resubmit our manuscript (RSOS-200584.R1), entitled ‘ Optimization of nitrogen and carbon removal with simultaneous partial nitrification, anammox, and denitrification (SNAD) in membrane bioreactor (MBR)’. We sincerely thank you and the reviewer for your valuable feedback that we have used to improve the quality of our manuscript. The reviewer comments are laid out below in italicized font and specific concerns have been numbered. Our response is given in normal font and changes/additions to the manuscript are given in blue text.

We hope that the revised version of the manuscript could be considered for publication in your journal. I look forward to hearing from you soon.

With best wishes,

Yours sincerely,

Kai Zhang,

Jul 28, 2020

To reviewer:

We sincerely thank you for your professional review work on our manuscript. Your thoughtful comments and constructive suggestions have contributed a lot to improve the quality of our manuscript. As you are concerned, there are several problems that need to be addressed. According to your nice suggestions, we have made extensive corrections to our previous draft. After this revision, we have written a point-by-point response letter to you as you can see above. And the detailed corrections are listed below.

Reviewer 2

Comment 1: *Still not clear to me how the experimental results exhibited in Table 2S were obtained. During the operational period the conditions tested are not the same as in table 2S. How were these experiments done? When DO concentration of 1.3 were tested? How accurate was it to keep DO levels at 0.3, 0.8 and 1.3?*

Response: We sincerely thank you for your professional review work on our manuscript. We are very sorry that we did not clearly answer your questions in the last revision. During the experimental design process (optimal operating conditions), C/N, DO, and Tae were selected as the three independent effective variables. According to the values of three independent variables, 17 operating parameters under different conditions were obtained by RSM, and then batch tests were carried out according to different operating parameters (Three replicate tests were conducted), so as to obtain the removal rates of different pollutants.

During the long operation of the reactor, as the reviewer raised, it was difficult for DO to maintain a constant value. Therefore, in the experiment, we adjusted DO in the reactor according to the change of effluent water quality by adopting low aeration rate and intermittent aeration. However, there is no corresponding relationship between DO value in long-term operation and DO value in supplementary Table 2S. The experimental data in supplementary Table 2S are the results of independent experimental designs, and the optimal reaction conditions are obtained through batch tests.

According to the suggestions of the reviewer, we have supplemented the periodic DO changes during the long-term operation of the reactor in **Comment 4**. During the batch test, under the condition of WTW real-time monitoring, the DO in the beaker was real-time regulated by the aerator, so that the DO value was controlled around the predetermined value. We hope our response can be up to your standard.

Comment 2: *All operational changes are still being difficult to follow. If it is difficult to indicate them in the figure, you can try to indicate in table 1 with more rows indicating what the conditions were and their respective period of days.*

Response: We sincerely thank you for your professional review work. We have tried to add more rows in Table 1 to show the change pattern of DO at each phase. However, when we tried to add it, we found that the reactor was not a stable DO concentration due to the intermittent aeration method we adopted, and it was difficult to visually describe the operation in Table 1. Therefore, according to your suggestion, we adopted the form of boxplot to conduct statistical analysis of the DO concentration at each phase of the non-aeration stage and the aeration stage in the long-term operation process. As shown in **Fig.3S** supplement. We monitored the DO concentration under intermittent aeration for two consecutive cycles in each phase. From the monitoring data, it can be

found that in each phase, the DO concentration in the non-aeration stage and the aeration stage can be basically kept in a stable range, and a certain DO concentration gradient is formed, which provides necessary conditions for the micro-environment in the reactor. We hope our response can be up to your standard.

Comment 3: *There are some conditions where the TIN removal efficiency was less than 70% while the graph in figure 2 the measured efficiencies start at 75%.*

Response: Thank you for your careful check. Fig.2 is obtained by software simulation based on the data in supplementary Table 2S. However, as mentioned by the reviewer, the TIN removal rate under some conditions was lower than 70%, which was the result of analysis under the condition of long-term operation of the reactor, and there was no corresponding relationship between the results and the data in Fig.2.

Fig.2, supplementary Fig.1S and supplementary Table 2S are the experimental results and optimal experimental conditions obtained through a series of batch tests using the response surface method. Finally, the reliability of the optimal condition is analyzed through the operation results of supplementary Fig.2S. We hope our response can be up to your standard.

Comment 4: *Information should be added about the periodicity of dissolved oxygen measurements, and the variability of monitoring, since it is difficult to maintain constant low oxygen concentrations with aeration systems and this was an important variable in the process.*

Response: We sincerely thank you for your professional review work on our manuscript. According to your thoughtful suggestion, we have supplemented the changes of the monitoring values of DO in the non-aeration stage and aeration stage in the long-term operation phase. As shown in supplementary Fig.3S. In the long-term operation phase, since the aeration mode of each phase is intermittent aeration, the changes of DO in two consecutive groups (non-aeration + aeration) are respectively monitored in each phase.

Because the reactor uses intermittent aeration, DO is difficult to maintain a constant value. Therefore, by adjusting the non-aeration time and the aeration time, a relatively stable DO gradient difference was formed inside the reactor. In order to more intuitively represent the DO values in the non-aeration stage and the aeration stage in the reactor, boxplot was adopted in supplementary Fig.3S for display. We hope our response can be up to your standard.

Fig.3S. The periodic change of DO concentration in different phases.

Reviewer 3

Comment1: *The authors have made some modifications and the responses are appreciated. The only reminder is that some recent publication on this topic should be updated.*

Response: We sincerely thank you for your thoughtful suggestion on our manuscript. According to your suggestion, we have reviewed the latest literature and updated the corresponding references in the manuscript. In the revised manuscript we have marked these changes in blue. We hope our

revised manuscript can be up to your standard.